# Comparison of Tensile Properties of Glass Fibre Reinforced Polymer Rebars by Testing According to Various Standards

**DOI:** 10.3390/ma13184110

**Published:** 2020-09-16

**Authors:** Agnieszka Wiater, Tomasz Siwowski

**Affiliations:** Department of Roads & Bridges, Rzeszow University of Technology, Al. Powstancow Warszawy 12, PL–35959 Rzeszow, Poland; siwowski@prz.edu.pl

**Keywords:** GFRP rebars, tensile properties, tensile testing, test standards

## Abstract

The widespread use of glass fibre reinforced polymer (GFRP) bars in reinforced concrete (RC) elements has yet been limited due to the anisotropic and non-homogeneous material behaviour of GFRP. The material characteristics of GFRP bars from different manufacturers vary as a function of several factors. Several standards have developed various procedures to investigate the mechanical characteristics of GFRP bars, but universal methods to test different types and diameters of GFRP bars in tension have not been fully developed. Due to the lack of such a standardized test procedure, there are some doubts and gaps in terms of the behaviour of GFRP bars in tension, which has led to lack of reliable information on their tensile properties. The determination of tensile characteristics of GFRP bars, including the tensile strength, modulus of elasticity, and ultimate strain, according to various test standards, is the main subject of the paper. This paper reports test results for tensile characterization obtained on four types of GFRP bars from four manufacturers with six various diameters. Moreover, the study compares various test procedures according to seven standards to characterize the tensile properties of GFRP bars, to examine the proposed test procedures, and to reveal main differences.

## 1. Introduction

Glass fibre reinforced polymer (GFRP) reinforcement has emerged as an attractive alternative of steel rebars due to the higher ultimate tensile strength to weight ratio, resistance to corrosion and chemical attack, electromagnetic neutrality, and long-term strength and durability of GFRP bars in harsh and corrosive environments [1,2,3,4]. GFRP bars are mainly used in the construction or rehabilitation of bridges [5,6,7,8,9,10,11]. Enormous potential exists for applications in multi-story buildings, parking garages, and industrial structures. In these scenarios, GFRP could replace the steel bars in concrete susceptible to corrosion. 

However, the widespread use of GFRP bars in reinforced concrete (RC) elements has been limited due to the anisotropic and non-homogeneous material behaviour of GFRP. Material characteristics of GFRP bars from different manufacturers vary as a function of several factors, including the type and proportions of fibre and resin and the quality control of the final products. Moreover, the difference in geometric dimensions and reinforcement ratio over the cross-section of the bars entails different resistance to applied loads. GFRP rebars may also differ in both raw components and surface texture (projection), which governs their mechanical behaviour. These factors might constitute an effective parameter in the diversity of the reported values for the strength and modulus of elasticity of GFRP bars.

There have been multiple studies on the characterization of GFRP bars in tension [12,13,14,15,16,17,18,19]. Given the fact that GFRP reinforcing bars cannot resist the required lateral confining pressure from mechanical grips during the material tests, the test method for GFRP bars needs to be different from the conventional steel bars. Several standards and codes have developed various procedures to investigate the mechanical characteristics of GFRP bars [20,21,22,23,24,25,26,27]. Despite numerous studies, universal methods to test different types and diameters of GFRP bars in tension have not been fully developed. Existing test standards differentiate the determination of bar diameter and tensile properties as well as test requirements and procedures. However, GFRP bars should be tested using a universal method so that the various products on the market could be examined similarly. The knowledge of these characteristics assists the analysis and design of structural concrete members reinforced with GFRP bars which are under flexure and tension.

The determination of tensile characteristics of GFRP bars, including the tensile strength, modulus of elasticity, and ultimate strain, according to various test standards, is the main subject of the current paper. This paper reports test results for tensile characterization obtained on four types of GFRP bars from four manufacturers, with six various diameters and two manufacturing factors. At least three specimens of each type/diameter/factor were tested to yield a statistically valid mean and distribution of tensile strength. 55 bar specimens were investigated. The data obtained through the tensile tests were analysed using a statistical data analysis software program, Matlab. Moreover, an investigation of load–strain behaviour and failure modes, and their correlation to the tensile characteristics are discussed.

## 2. Research Significance 

There is a fast-growing number of GFRP bars used in construction, and many variables determine the material characteristics of these products. Because of this, a comprehensive investigation of the most recent GFRP reinforcing bars was believed to be an essential step toward designing GFRP reinforced concrete structures, demonstrating satisfactory performance. Various mechanical characteristics have been reported for GFRP bars in tension by researchers using different test procedures. Due to the lack of such a standardized test procedure, there are some doubts and gaps in terms of the behaviour of GFRP bars in tension which led to a lack of reliable information on their tensile properties, i.e., tensile strength, modulus of elasticity, and ultimate strain. Moreover, the notable scatter in the gathered results occurred because of the various bars’ diameters, types of resin used in manufacturing, as well as fibre volume content of the tested FRP bars. Therefore, this study compares various test procedures according to standards [20,21,22,23,24,25,26,27] to characterize the tensile behaviour of GFRP bars and presents an experimental study to examine the proposed test procedures and reveal the main differences. The influence of bar type (manufacturer), diameter, and some manufacturing factors on the tensile characteristics of tested GFRP bars is also presented in the paper. Because of the significant differences in tensile properties revealed in this research, the necessity of test standards unification is realized. 

## 3. Materials and Methods

### 3.1. GFRP Bars

The GFRP bars were produced by the pultrusion process and comprised continuous glass fibres with different fibre contents bonded together with epoxy or vinyl ester resin. In the current research GFRP bars from four different manufacturers (Figure 1) with size (diameter) range from 5 mm to 12 mm were tested. A total of 12 series of tests were performed, with at least three replicate specimens tested for each of them (Table 1). The series’ designation of tested bars (e.g., B–D5c) uses the letter “A”, “B”, “C”, and “D”, referring to the type of bars (manufacturer). The next letter “D” preceding the number indicates the producer’s designation of the size of the bar, e.g., D8 means 8 mm diameter of the bar (interior or exterior, it depends on manufacturer). The additional letters “a”, “b”, or “c” refer to various manufacturing batch and/or resin modification used in manufacturing. The fibre content of bars was about 75%–80% by weight for A and B types and over 80% for C and D types. Bars of A and B types have the spiral braid of fibres around the bars, bars of C type have a sand coating, while D bars have a ribbed surface that is cut into the hardened bars. In the last case, the ribs do not take part in the transfer of tension load.

### 3.2. Effective Bar Diameter

To calculate the tensile properties, the diameter of the bar must be known. Most of the standards [21,22,23,25,26,27] recommend using an underwater weighting method to determine the diameter of the bar. Only standard [24] prescribes measurement of the diameter with a micrometer and the mean of six readings to take into account. The required accuracy of measurements should be 0.015 for diameter in the range of 3–10 mm and 0.050 for diameter in the range of 10–20 mm. 

In the current research, both methods of determining bars’ diameter were used: underwater weighing (3 measurements) and measuring by micrometre with appropriate accuracy (6 measurements). No attempt was made to determine the diameter of each bar. Instead, the average diameter was determined from a randomly chosen bar of each series. The average value obtained by underwater weighting is designated as “effective diameter”. In this study, due to the assumption that any ribs and sand coating had not taken part in carrying tension stress, only solid bar cores were taken into further analysis and designated as “core diameter”. Therefore, in the case of the sand-coated bars, the sand layer was carefully removed before measurement. In the case of series of D type bars, to determine the core diameter, twice the hight of the ribs, measured with a micrometer, is subtracted from the exterior diameter. 

The average diameter values of each series are presented in Table 2. The difference between “core” and “effective” diameters and relevant cross–sectional areas are in the range of 1.8%–7.2% and 3.6%–13.9%, respectively. This brings in considerable differences to the calculation of tensile strength and modulus of elasticity through bar cross–sectional area. Moreover, the diameter measurement results for the bars with the same size (12 mm) but four various surface treatments differ by up 18.6% and 22.5% deviation for core and effective diameter, respectively. Subsequently, this causes the cross-sectional areas of bars with the same size to differ by up to 33.7% and 40.0% deviation for core and effective diameter, respectively. It is worth noting that smaller deviations are obtained for core diameter measurement. 

### 3.3. Specimen Preparation

The use of conventional grips for testing unidirectional pultruded bar having high axial tensile strength and low transverse compressive strength can cause premature failure of the bar due to high compressive force applied by the grips. The special anchorages should be used to reduce the compressive forces exerted on the bar. The anchorages shall be suited to the geometry of the test rebar and must allow the full strength of the bar to be obtained. Furthermore, anchors shall ensure concentric and torsion-free loading and shall ensure that specimens rupture occurs outside the anchorage zone. Most standards recommend anchorages of steel tubes filled with expansive cement grout, polymer resin, or 1:1 mixture by weight of resin and sand. However, anchorages are recommended but not required, and alternative methods for attaching specimens to the testing machine are acceptable. For example, steel tabs adapted to the shape of the bar are recommended by standards [21,24]. 

The total length of the specimen shall be taken to be the sum of the length of the test section and the length of the anchoring section or greater. The requirements for the free length between the anchorages depend on the test standard. Most of them recommend that the length of the test section shall be not less than 40 times the effective bar diameter. The detailed requirements of the standards are presented in Table 3.

In this research, the ends of GFRP bar specimens were embedded into steel tubes filled with expansive cement grout. The PCV cups with a concentric through–hole of appropriate diameter were used for plugging steel tubes and centring a bar inside the tube (Figure 2). This facilitated gripping of the bar specimens and prevented the premature failure of bars at steel jaws of the testing machine. The length between anchorages was taken as follows: 380 mm for diameters 5mm and 6 mm, 400 mm for diameter 8 mm and 10 mm and 480 mm for 12 mm diameter. Thus, the requirements of all considered standards were met.

### 3.4. Test Procedure

Specimens shall be instrumented with longitudinal strain gauges or extensometers to measure the elongation of the test piece under loading. The gauge length of the extensometer shall be met requirements according to the appropriate standard presented in Table 3. Most of the standards required a gauge length not less than eight times the nominal diameter of the tested bar. The extensometer shall be placed at in the centre of the test section. In this research gauge length of an extensometer was 100 mm and met the requirements listed in Table 3. 

The tensile load shall be applied at a constant rate until the failure of the specimen. Most of the standards [21,23,24,25,26] recommend a constant strain rate and determines the duration of the experiment. Other standards [22,27] propose the constant stress rate. The detailed requirements of the standards in terms of test duration and speed are presented in Table 3. In this research, bars were loaded at the constant strain rate 4 mm/min regardless of bar diameter. This resulted in the fact that specimens failed after 3–7 min of loading.

All tensile tests were carried out using a 1200 kN—capacity universal testing machine with video non-contacting extensometer (Figure 2). The tests of bars within each group were run consecutively but in random order. Since all bars in a group were tested within a short time (1 h), it was assumed that the test conditions within a group were similar. Also, for each group, a single batch of cementitious grout for gripping was prepared, thus eliminating any variation due to grout differences among the bars within each group. These measures reduced resultant variations due to changes in test conditions over time.

## 4. Test Results

### 4.1. Load-Strain Behaviour

The observed tensile Load-strain behaviour of tested GFRP bars is presented in Figure 3, Figure 4 and Figure 5. This was to be expected that all specimens showed linear behaviour up to failure. Moreover, all specimens in particular series failed at similar load level, but sometimes at the different ultimate strain. The greatest relative scatter of ultimate strains is observed for all A type specimens and B type specimens with 5 mm and 6 mm diameters. It could show the worst manufacturing quality (in terms of tensile properties homogeneity) for A and B manufacturers—the latter is only concerned with bars with the smallest diameters (5 mm and 6 mm). However, since the perfect parallelism in the load–strain curves is observed for almost all specimens, the modulus of elasticity seems to be constant, with no considerable scatter for all bar types or diameters. Some linearity disturbance observed at an early stage of the test for A series of specimens is due to an initial adjustment of the extensometer.

### 4.2. Failure Modes

All specimens failed suddenly as expected. The observed tensile failure mode of a representative specimen of each type of tested GFRP bars are shown in Figure 6. All the failures started with splitting (fibres delamination) and ended with rupture of the bar due to tensile rupture of fibres. Every bar split into different numbers and sizes of pieces. It can be noted that the A and B types of bars indicated less extensive mode of failure. This might be related to the type of surface treatment. The A and B type bars had continuous spiral ribbing made of fibres wounded around the bars which hold the longitudinal fibres and prevent spreading out. On the other hand, the C and D types indicated higher tensile strength and modulus of elasticity than A and B types what also might be related to the failure mode of bars. All the failures occurred within the test section. This shows that the selected gripping method was sufficient to prevent uncontrolled slippage of bars.

### 4.3. Tensile Strength

Tensile strength varies with the cross–sectional dimensions of the bar due to shear lag effect and therefore tensile strength should be referenced based on the bar size. In case of all standards [21,22,23,24,25,26,27], in order to determine the tensile strength, the highest recorded load shall be divided by the cross-sectional area according to Equation (1): (1)$$f_{u}\;=\frac{F_{u}}{A}\;\left[\mathrm{Mpa}\right]$$
where $F_{u}$ is the maximum tensile force (N) and A the cross–sectional area (mm^2^).

However, the cross-sectional area of each bar depends on the bar diameter, which is determined as “core” and “effective” one (Table 2). In Figure 7, the results of individual tests and the average value of tensile strength for each series based on core and effective diameters are plotted. For core diameters, the determined values are in the range of 1031 MPa–1415 MPa. The highest tensile strength was obtained for C–D12 series while the lowest value was obtained for B–D5c series of tested bars. Test results indicate that types A, C and D exhibited similar results while type B is shown to have a mean tensile strength significantly different from the other three. This difference might be related to the volume fraction of fibres, which is the least for B type bars. Moreover, the series C and D indicate the tensile strength over 1200 MPa. This is due to the higher fibre content in comparison of A and B series. The slight influence of bar diameter on tensile strength is also observed. The strength for 10- and 12-mm bars are higher in comparison to the ones with a smaller diameter. This is observed for A, C and D types, while the B type bars are more homogeneous in terms of tensile strength despite the bar diameter. The method of measuring the diameter affects the tensile strength values, which are proportionally smaller for effective diameter. Since only standard [24] prescribes measurement of the core diameter with a micrometer, the tensile strength determined according to this standard is about 4%–16% higher depending on bar type and diameter (Table 4). 

Statistical data analysis of tensile strength test results is reported in Table 4. The coefficient of variation is highest for B–D5a (8.5%) and B–D11 (5.35%) series, while for the rest series its value is generally less than 4%. Such a high variability for both B series could mean that the average strength in these cases would not provide a reliable estimate of the true mean strength. However, the lowest coefficient of variation was obtained for B–D5b series (0.95%), which can indicate the influence of manufacturing factors on tensile strength variability. The method of measuring the diameter does not affect the variability of the tensile strength values. 

### 4.4. Modulus of Elasticity

Modulus of elasticity is the slope of the load–strain curve obtained from extensometer or strain gauge readings. Equation (2) shall be used to calculate the value of the modulus of elasticity:(2)$$E\;=\frac{\Delta \sigma }{\Delta \varepsilon }\;\left[\mathrm{Mpa}\right]$$
where Δσ denotes the difference in applied tensile stress between the starting and ending points (MPa) and Δε the difference in tensile strain between the starting and ending points.

The start and end points of the load–strain curve chosen to modulus calculation depend on the standards used. Some standards specify strain interval [16,19] and others specify stress levels [22,23,25,26,27], which shall be used to modulus calculation. The detailed specifications of the standards are presented in Table 5.

The calculation of modulus of elasticity was made according to all presented standards. If data for the exact stress/strain point were not available, the closest available data were applied. In Figure 8a–e the results of individual tests and average values for each series are plotted. A statistical data analysis (mean value (AVG), standard deviation (SD) and coefficient of variation (CV)) of test results is reported for each series in Table 6 as calculated according to all standards. 

The average modulus of elasticity ranges from 41 to 68 GPa when based on core diameter and from 38 to 61 GPa when based on effective diameter (i.e., 11% less). The lowest values of moduli were obtained for A–D8a and B–D5a series and the highest value for C–D12 series, regardless of the standard used for determination. Modulus of elasticity is affected by the level of fibre content [12]. The higher values of modulus of elasticity (above 60 GPa) were obtained for the C and D series when compared to A and B series due to the higher fibre content. The slight influence of bar diameter on the modulus of elasticity is observed in A series only. The modulus increases with diameter. However, this relationship is not observed in the B series. The method of measuring the diameter affects the modulus of elasticity values, which are proportionally smaller for effective diameter. The same as for the tensile strength, the modulus of elasticity determined according to standard [24] is about 1%–18% higher depending on bar type and diameter (Table 6). The highest values are obtained for bars with the greatest diameter and according to standard [27].

Statistical data analysis of modulus of elasticity test results is reported in Table 6. The 3–4 times greater scatter of moduli values was observed for determination based on standards [21,24]. In these cases, the coefficients of variation were in the range of 4%–31% and 2%–24% for both standards, respectively. The greatest values were obtained mainly for A series. Such a high variability for A series could mean that the average modulus of elasticity in these series would not provide a reliable estimate of the true mean modulus. However, these both standards specify strain interval for modulus calculation at a very early stage of the test; the end point is required for a strain of 0.003 and 0.005, respectively. The greater coefficient of variation in case of standards [21,24] is due to extensometer adjustment and linearity disorder observed at the early stage of the tensile test. The test reveals that calculation of modulus of elasticity based on measurement in the range of 20%–60% of the ultimate load seems to be the more reliable method because, in this case, a much smaller scatter of modulus was obtained. According to standards based on this assumption the coefficient of variation is highest for A-D8a (7.02%) and A-D12 (6.83%) series, while for the remaining series, its value is generally less than 4%. The lowest coefficient of variation was obtained for B-D8 series (1.29%). The method of measuring the diameter does not affect the variability of the modulus of elasticity values. 

The analysis of test results carried out in this study reveals the high dependence of modulus of elasticity on the test standard used to determine its value. Two main requirements which distinguish these standards are: primarily, the location of start and end points of the load–strain curve taken into account in the modulus calculation and, secondly, the method of bar diameter determination (core or effective), which is used for Δσ calculation. To realize the reader how wide scatter of moduli values is, in Figure 9, Figure 10 and Figure 11, the average values of modulus are presented for all tested series, depending on the standard used for moduli determination.

### 4.5. Ultimate Strain

The most of standards [21,22,24,26,27] state that the ultimate strain determined in tensile tests shall be a strain corresponding to tensile strength (in this paper called as “Method I”). It is possible to determine when strain measurement is possible up to failure. The alternative method of ultimate strain determination (in this paper called as “Method II”) is the calculation according to Equation (3), taking into account the tensile strength and modulus of elasticity: (3)$$\varepsilon_{u}\;=\frac{f_{u}}{E}=\frac{F_{u}}{A\cdot E}\;\left[-\right]$$
where $F_{u}$ denotes the maximum tensile force (N), A is cross-sectional area (mm^2^), and E is modulus of elasticity (MPa).

Method II is also acceptable by most of the standards [22,23,25,26,27]. In the case of this method, the modulus of elasticity determined according to standard [22] was taken into consideration, because this standard allows using both methods for determining the ultimate strain.

In Figure 12, the results of individual tests and the average value of ultimate strain for each series obtained with both methods are plotted. The average value of ultimate strain ranges from 2.03 to 2.46 for Method I and from 1.97 to 2.52 for Method II. In the case of both methods, the lowest value of ultimate strain was obtained for B-D5c series and the highest value was obtained for A-D8a series. It should be noted that, for B-D5c series, the lowest tensile strength was obtained as well. The influence of fibre content and bar diameter on ultimate strain was not observed. The method of ultimate strain determination has a slight effect on its average value. This effect can be measured in the percentage difference estimated in the range from −4.63% to 3.04% (Table 7). 

Statistical data analysis of ultimate strain test results is reported in Table 7. In the case of Method I, the coefficient of variation ranges from 0.84% to 16.54%, while for Method II its value is much smaller and ranges from 1.50% to 6.41%. The coefficient of variation is highest for A-B8a series (16.54%/6.41%). Such a high variability for all A series could mean that the average ultimate strain in these cases would not provide a reliable estimate of the true mean strain. The lowest variability showed C-D12 and D-D12 series (0.93% and 0.84%, respectively), but for Method I only. In the case of B series, the highest variability was revealed for B-D5a series (8.67%/6.02%), which can indicate the influence of manufacturing factors on ultimate strain variability. Generally speaking, the method of determination has a quite significant effect on the variability of the ultimate strain values. Since Method II shows less scattered values, it seems to be more appropriate for determination of ultimate strain.

## 5. Discussion

### 5.1. Comparison of Tensile Properties of Bar Types

To compare the tensile properties of GFRP bars produced by four different manufacturers, the series with the same diameter (12 mm) were taken into consideration, i.e., A–D12, B–D11, C–D12, and D–D12. This size of GFRP bars is also the most commonly used in construction (f.e., bridge deck slabs [5,6,7,9]). The tensile properties of four type of bars are listed in Table 8. The properties gathered in this table are determined as follows: tensile strength is based on core diameter, modulus of elasticity is calculated according to standard [22] and ultimate strain is obtained by “Method II”.

As far as the tensile strength and modulus of elasticity are concerned, the highest values for both were determined for C type bars, while the lowest for B type bars. The difference between these two types of bars for both parameters is about 20%. It is mainly due to fibre content; lowest for B series, much greater for C series (Table 1). However, the case of D type bars, with the highest fibre content but medium tensile properties, indicates that not only this feature is decisive. The tensile properties of A and D type bars were at a similar level. The greatest scatter of tensile strength was obtained for B type (>5%) and modulus for A series (>7%). The average ultimate strain was at a similar level for all types; the biggest difference was only 2.7%. For this property, the greatest scatter was obtained for D type (>5%).

### 5.2. Influence of Bar Diameter on Tensile Properties

Influence of bar diameter on tensile properties was evaluated for two bar types (A, B) for which bars with at least three various diameters were tested. There are A–D8a, A–D10, A–D12 series for A type and B–D5a, B–D8, and B–D11 series for B type. Tensile properties considered in this analysis were determined in the same manner as previously (Section 5.1). The values obtained the same method as in the previous chapter was adopted to the analysis. Figure 13, Figure 14 and Figure 15 present the influence of bar diameter on tensile strength, modulus of elasticity and ultimate strain of A and B bar types, respectively. The standard deviations, as well as the trend lines (based on linear regression), were also shown in the following plots.

No clear influence of bar diameter on the tensile properties of GFRP bare was determined in the current research. There were only three diameters taken into consideration and it seems to be not sufficient to evaluate the objective influence of bar diameter on its tensile properties. Therefore only slight relationships based on the limited number of tests can be discussed as follows. In case of both type of bars, the increase of bar diameter up to 8 mm increased tensile strength and modulus of elasticity, but further increasing of diameter showed the opposite effect. However, the trend lines show the proportional increase of both tensile properties for A type and constant values for B type. This is a bit surprising because the tensile strength of GFRP bars generally tends to reduce with an increasing diameter [1,2,3,4]. This could be explained by the fact that the stress developed at the fibres located near the surface is not fully transferred to the fibres located at the centre of the bar. The opposite impact of bar diameter on ultimate strain was observed; ultimate strain decreases with diameter increase and this reduction is much higher for A type than for B type of bars. Generally speaking, taking into consideration the slope of the trend lines, it can be noted that the tensile properties seem to be more sensitive to bar size (diameter) for A type of bars when compared with B type of bars. Moreover, the latter seems to be fully insensitive (i.e., constant).

### 5.3. Influence of Manufacturing Factors on Tensile Properties

The influence of two manufacturing factors: modification of resin and uniformity of batch on tensile properties was also investigated. To check if the tensile properties remain invariable regardless of a production batch, the A–D8a and A–D8b series were taken into account for the analysis, while the series of B–D5a, B–D5b, and B–D5c were compared to determine the impact of resin modification. The bars of B–D5a series were manufactured with the standard resin used by the company (reference series), while the B–D5b and B–D5c series had the modified resin in search of better technological parameters. The tensile properties of the above-mentioned series of bars are presented in Table 9.

The comparison of batch homogeneity for A type bars shows very good manufacturing quality. The tensile strength of A-D8a and A-D8b series shows very good compliance with the difference of less than 1%. The modulus of elasticity and ultimate strain of both series have also good agreement, however, the difference in these cases is bigger, about 4%–5%. Therefore, it can be said that the tensile properties of A type bars remain invariable between two manufacturing batches (means different manufacturing time). However, attention should be paid to the fact the A-D8a series has more than two times greater variability of test results. 

The modification of resin for B type bars had no significant influence on tensile strength and modulus of elasticity; the difference is in the range of 4%–9%. However, in the case of B-D5c series, the modification caused a significant impact on the ultimate strain, resulted in a 17% reduction. The resin modification had a positive influence on the homogeneity of test results. For all tensile properties, the coefficient of variation for B-D5b and B-D5c series was smaller than for the reference series even by nine times in the case of tensile strength.

### 5.4. Comparison of Test Standards

Unfortunately, there is no worldwide unified test standard for determination of tensile properties of GFRP rebars. It hinders to compare the products of various manufacturers and differentiates the design requirements (limit states) in various countries, where the national codes are mostly obliged. To evaluate the scale of this problem quantitatively seven test standards [16,17,18,19,20,21,22] were considered to determine of tensile properties of four types of GFRP rebars. In these standards, the following parameters/properties are calculated/determined in a different way: size (diameter) of the bar and thus cross-sectional area, tensile strength, modulus of elasticity, and ultimate strain. Moreover, the tensile test requirements as the length of the test section, gage length of extensometer and test duration and speed are also different in individual standards. Nevertheless, in the current study, the test requirements of all considered standards were met.

To establish the tensile properties of the bar, the diameter and subsequently the cross–sectional area should be first determined. In the current research two methods of diameter measurement were investigated: based on underwater weighting (“effective” diameter) and using a micrometer (“core” diameter). The difference between “core” and “effective” diameters and relevant cross–sectional areas are in the range of 1.8%–7.2% and 3.6%–13.9%, respectively. This brings in considerable differences to the calculation of tensile strength and modulus of elasticity through bar cross-sectional area. Only the European standard [24] requires to determine “core” diameter. However, it seems to be the only rational way, because any ribs and sand coating had not taken part in carrying tension stress. Therefore the solid bar core diameter and the relevant cross-sectional area should be taken into further analysis. 

There is no difference between standards for tensile strength calculation. However, the above–mentioned difference in diameter determination affects tensile strength values, which are proportionally greater for the “core” diameter. Thus the tensile strength determined according to standard [24] is about 4%–16% higher depending on bar type and diameter when compared to other standards.

In the case of modulus of elasticity determination, another difficulty precludes the direct standard comparison: the specification of how the start/end points at the load–strain curve should be considered for modulus calculation. The current test reveals that calculation of modulus of elasticity based on measurement in the range from 20% to 60% of ultimate tensile load [22,23,25,26,27] seems to be the more reliable method because, in this case, a much smaller scatter of modulus was obtained. The smallest scatter (0.73%–6.24%) was obtained for Canadian standard [22]. On the other hand, the same as for tensile strength, the modulus of elasticity determined according to standard [24] (i.e., for “core” diameter) is about 1%–18% higher depending on bar type and diameter. 

Two methods of ultimate strain determination were considered according to various standards: I, based on direct measurement, and II, based on calculation with previously determined tensile strength and modulus of elasticity. Since Method II shows less scattered values, it seems to be more appropriate for determination of ultimate strain. The advantage of this method is also no necessity to use extensometer up to the failure of the tested bar. Method II is also acceptable by most of the standards [22,23,25,26,27], unfortunately not by the European one [24]. 

Summing up, the current research reveals that for determination of GFRP bars’ tensile properties the following conditions should be taken into account: cross-sectional properties based on core diameter, the start/end points at the Load-strain curve in the range from 20% to 60% of the ultimate tensile load for modulus of elasticity determination, and Method II for ultimate strain determination. However, it does not exist such a test standard, including all these conditions. Hence, the paper reveals the main differences in test procedures according to various standards which can lead to adverse differentiation of the tensile properties of the same GFRP bars.

## 6. Summary and Conclusions

The paper reports the comprehensive tensile testing of four types of GFRP rebars (designated as A to D) to determine their tensile properties according to various test standards and to evaluate the type, size (diameter) and some manufacturing factors on tensile properties. More than 50 specimens of different type and size were tested and the results were evaluated according to seven standards.

The following conclusions can be drawn based on the test results of this research:Due to the existence of two standard methods for bar diameter determination, the difference between “core” and “effective” diameters and relevant cross-sectional areas are in the range of 1.8%–7.2% and 3.6%–13.9%, respectively. This brings in considerable differences to the calculation of tensile strength and modulus of elasticity through bar cross-sectional area.The rebars from different manufactures vary in the extent of tensile properties. The highest values of tensile strength and modulus of elasticity were determined for C type bars, while the lowest for B type bars. The difference between these two types of bars for both parameters is about 20%. It is mainly due to fibre content.No clear influence of bar diameter on the tensile properties of GFRP bare was determined. However, the trend lines show the proportional increase of tensile strength and modulus of elasticity for A type bars and constant values for B type, while the decreasing impact on ultimate strain was observed.Tensile properties of A type bars stay invariable in a different batch of manufacturing. The type of resin for B type bars had no significant impact on tensile properties but influenced data variation.For the determination of GFRP bars’ tensile properties, the following conditions should be taken into account: cross-sectional properties based on core diameter, the start/end points at the load–strain curve in the range of 20%–60% of the ultimate tensile load for modulus of elasticity determination, and Method II for ultimate strain determination.

Due to the irregular cross-sectional dimensions of the GFRP rebars, one should consider adopting standard limits for ultimate bar force instead of ultimate stress. Such an approach would eliminate the need to measure the effective bar diameter to demonstrate compliance with strength specifications. A core bar diameter could be used for the consideration of cover depth, development length, and bar spacing.

This study also established the variability of the tensile properties of GFRP bars produced by different manufacturers. It provides a better understanding of the inherent variability of GFRP reinforcement. Such information is critical for determining the number of required tests to demonstrate compliance with standards and to establish the characteristic (or minimum) properties for design purposes. The research results can help the construction industry to manage GFRP reinforcement implementation and thus to widespread their use for construction.

## Figures and Tables

**Figure 1 materials-13-04110-f001:**
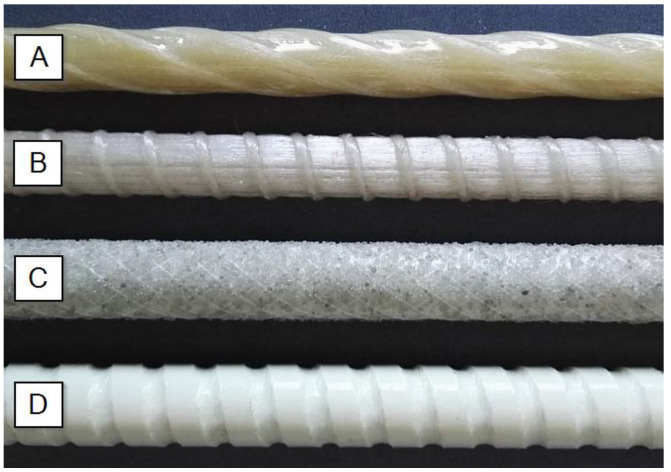
Types of tested GFRP bars. (**A**) AlbaKompozit; (**B**) ComRebars; (**C**) Pultrall; (**D**) Schoeck.

**Figure 2 materials-13-04110-f002:**
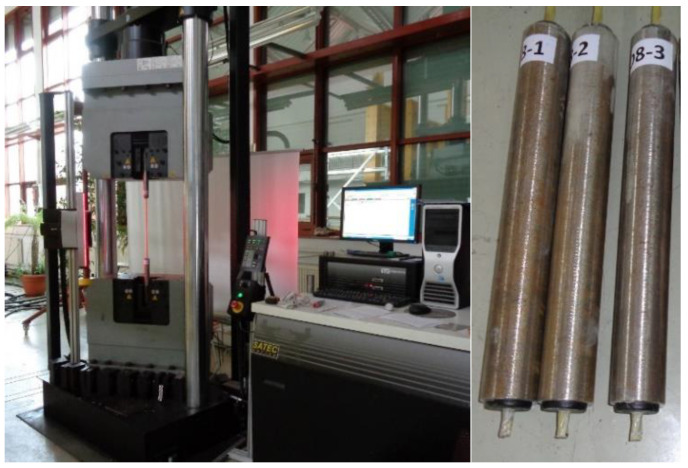
Exemplary GFRP bar gripped in the testing machine (left) and anchorages of bars (right).

**Figure 3 materials-13-04110-f003:**
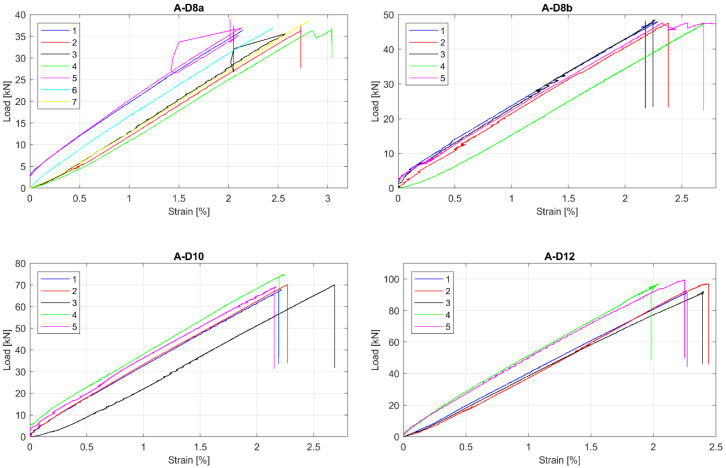
Load-strain curves for A series of tested bars. (**A-D8a**) type A, 8 mm, batch “a”; (**A-D8b**) type A, 8 mm, batch “b”; (**A-D10**) type A, 10 mm; (**A-D12**) type A, 12 mm.

**Figure 4 materials-13-04110-f004:**
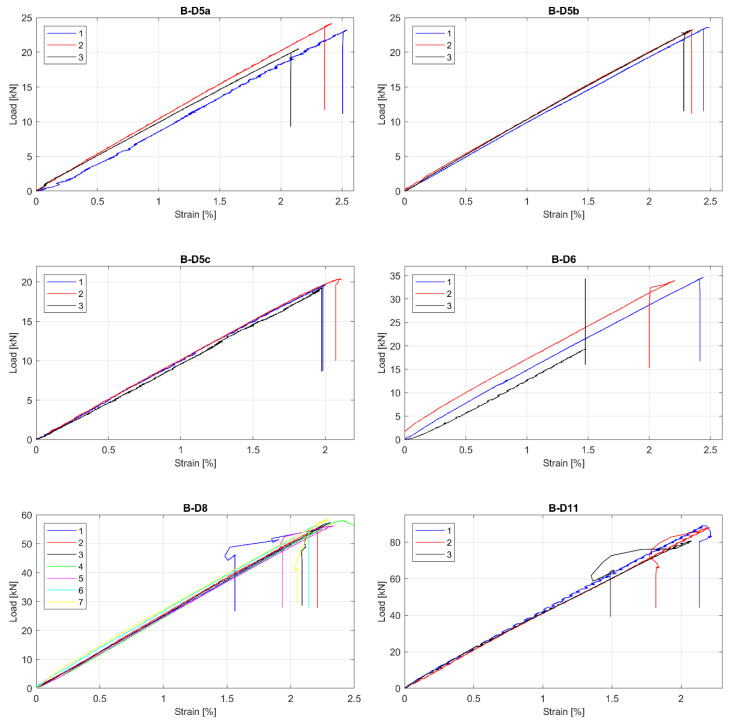
Load-strain curves for B series of tested bars. (**B****-****D5a**) type B, 5 mm, modification “a”; (**B-D5b**) type B, 5 mm, modification “b”; (**B-D5c**) type B, 5 mm, modification “c”; (**B-D6**) type B, 6 mm; (**B-D8**) type B, 8 mm; (**B-D11**) type B, 11 mm.

**Figure 5 materials-13-04110-f005:**
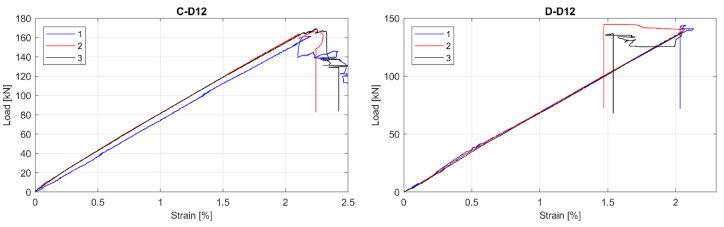
Load-strain curves for C and D series of tested bars. (**C-D12**) type B, 12 mm; (**D-D12**) type D, 12 mm.

**Figure 6 materials-13-04110-f006:**
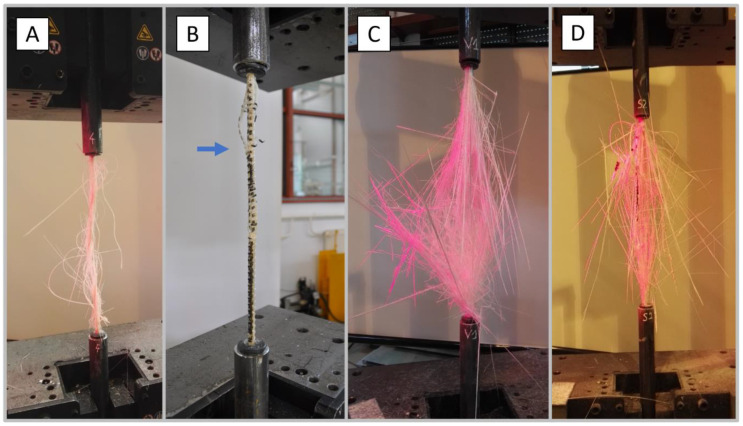
Typical failure modes of tested GFRP bars. (**A**) AlbaKompozit; (**B**) ComRebars; (**C**) Pultrall; (**D**) Schoeck.

**Figure 7 materials-13-04110-f007:**
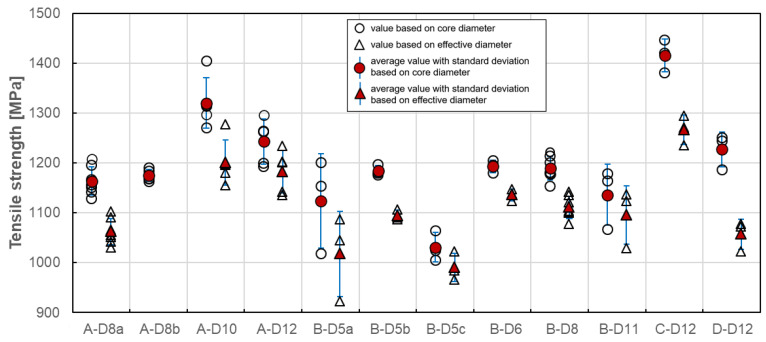
Tensile strength of tested GFRP bars.

**Figure 8 materials-13-04110-f008:**
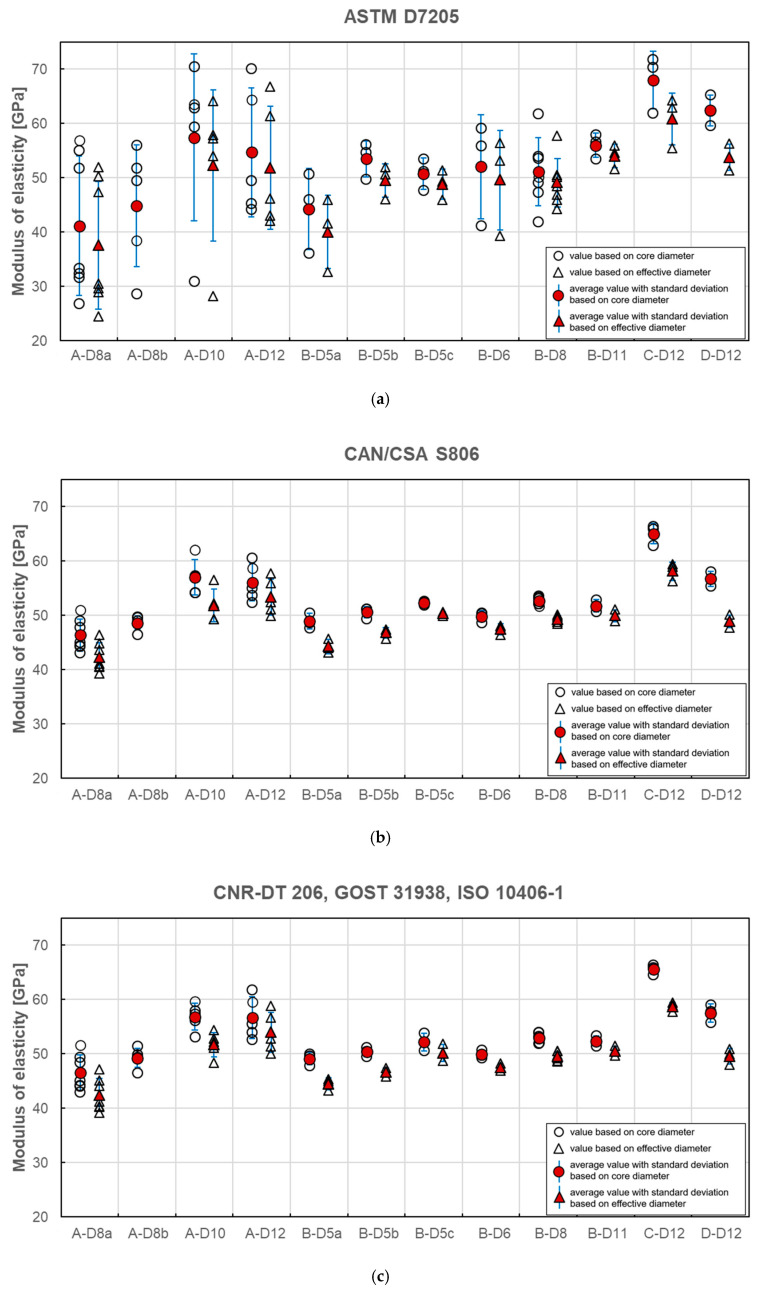
Modulus of elasticity obtained according to standard (**a**) ASTM D7205; (**b**) CAN/CSA S806; (**c**) CNR-DT 206; GOST 31938, ISO 10406-1; (**d**) EN 17129; (**e**) JSCE-E 531 [21,22,23,24,25,26,27].

**Figure 9 materials-13-04110-f009:**
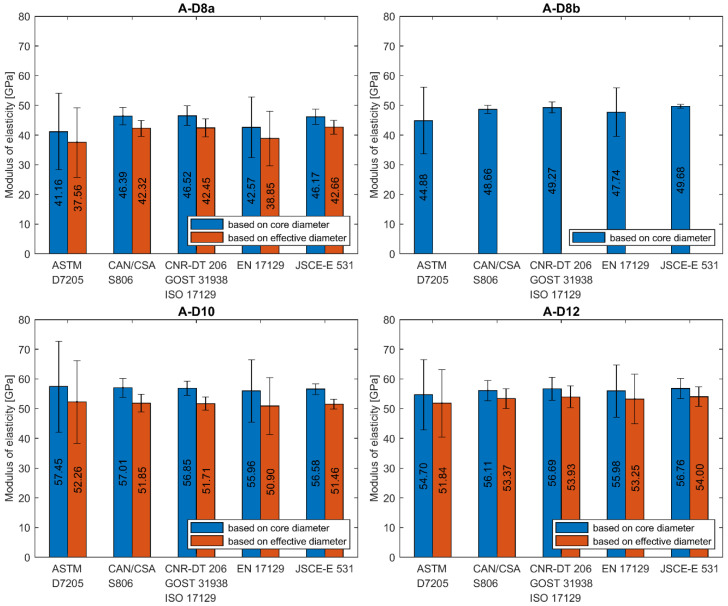
Modulus of elasticity of A series’ bars determined according to different standards. (**A****-****D8a**) type A, 8 mm, batch “a”; (**A-D8b**) type A, 8 mm, batch “b”; (**A-D10**) type A, 10 mm; (**A-D12**) type A, 12 mm.

**Figure 10 materials-13-04110-f010:**
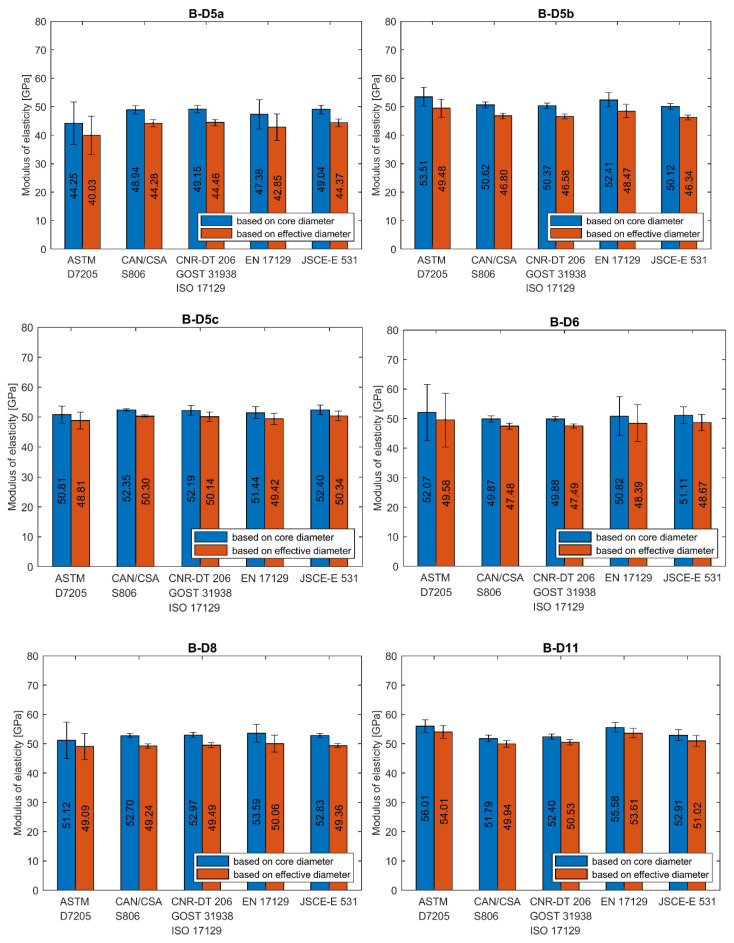
Modulus of elasticity of B series’ bars determined according to different standards. (**B****-****D5a**) type B, 5 mm, modification “a”; (**B-D5b**) type B, 5 mm, modification “b”; (**B-D5c**) type B, 5 mm, modification “c”; (**B-D6**) type B, 6 mm; (**B-D8**) type B, 8 mm; (**B-D11**) type B, 11 mm.

**Figure 11 materials-13-04110-f011:**
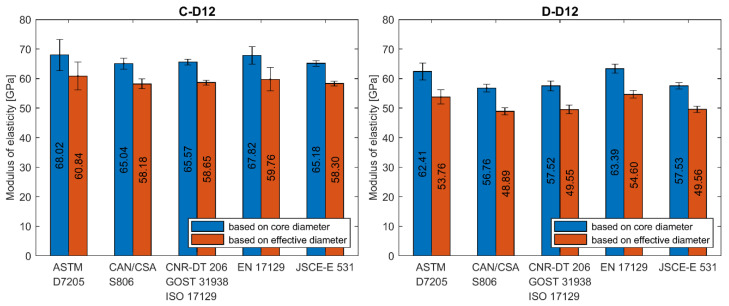
Modulus of elasticity of C and D series’ bars determined according to different standards. (**C-D12**) type C, 12 mm; (**D-D12**) type D, 12 mm.

**Figure 12 materials-13-04110-f012:**
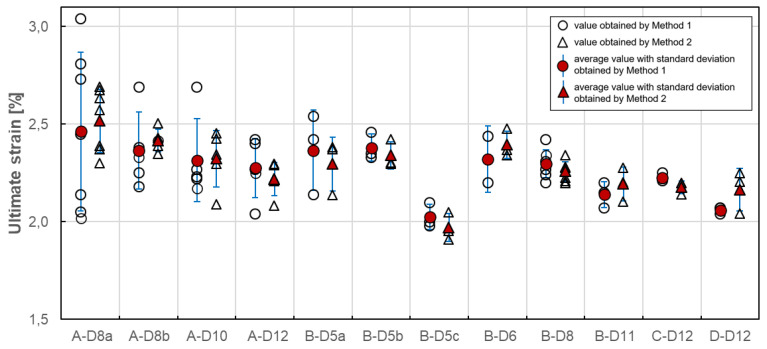
Ultimate strain of tested GFRP bars.

**Figure 13 materials-13-04110-f013:**
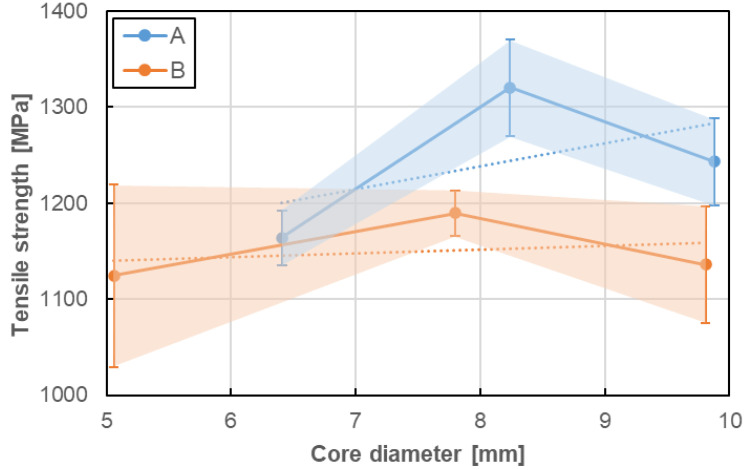
Influence of bar diameter on tensile strength of A and B bar types.

**Figure 14 materials-13-04110-f014:**
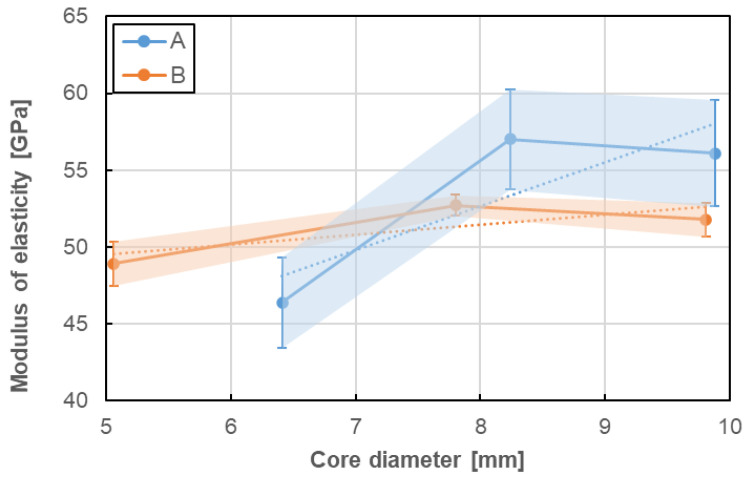
Influence of bar diameter on modulus of elasticity of A and B bar types.

**Figure 15 materials-13-04110-f015:**
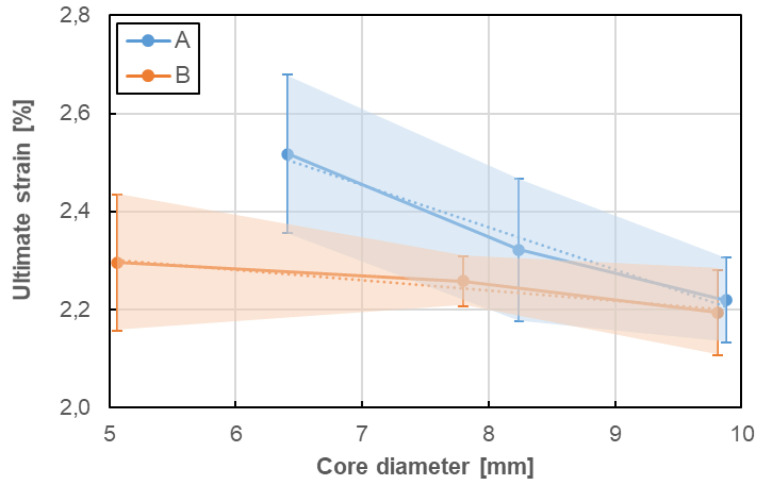
Influence of bar diameter on the ultimate strain of A and B bar types.

**Table 1 materials-13-04110-t001:** Details of tested GFRP bars.

Type	Resin	Fibre Content by Weight [%]	Surface Treatment	Bar Size [mm]	Manufacturer
A	epoxy	77.5	continuous spiral ribbing	8, 10, 12	AlbaKompozit https://www.albakompozit.pl/
B	epoxy	75.0	continuous spiral ribbing	5, 6, 8, 11	ComRebars https://www.comrebars.eu/
C	vinyl ester	83	sand–coated	12	Pultrall (V–Rod) https://www.fiberglassrebar.com/
D	vinyl ester	88	ribs cut into hardened bars	12	Schoeck https://www.schoeck.com/

**Table 2 materials-13-04110-t002:** Geometric characteristics of tested bars.

Title	Core Diameter [mm]	Core Area [mm^2^]	Effective Diameter [mm]	Effective Bar Area [mm^2^]	Diameter Difference [%]	Area Difference [%]
A–D8a	6.41 ± 0.26	32.27	6.71 ± 0.11	35.36	4.47	8.74
A–D8b	7.20 ± 0.35	40.72	n/a	n/a	n/a	n/a
A–D10	8.24 ± 0.06	53.33	8.64 ± 0.05	58.63	4.63	9.04
A–D12	9.88 ± 0.04	76.67	10.13 ± 0.02	80.60	2.47	4.88
B–D5a	5.06 ± 0.05	20.11	5.32 ± 0.05	22.23	4.89	9.54
B–D5b	5.01 ± 0.18	19.71	5.21 ± 0.03	21.32	3.84	7.55
B–D5c	4.94 ± 0.20	19.17	5.04 ± 0.04	19.95	1.98	3.91
B–D6	6.05 ± 0.11	28.75	6.20 ± 0.03	30.19	2.42	4.77
B–D8	7.80 ± 0.06	47.78	8.07 ± 0.04	51.15	3.35	6.59
B–D11	9.81 ± 0.44	75.58	9.99 ± 0.07	78.38	1.80	3.57
C–D12	12.20 ± 0.05	116.90	12.90 ± 0.06	130.70	5.43	10.56
D–D12	12.14 ± 0.05	115.75	13.08 ± 0.09	134.37	7.19	13.86

**Table 3 materials-13-04110-t003:** Tensile test requirements in various standards.

Test Standard	Length of Test Section	Gage Length of Extensometer	Test Duration and Speed
ASTM D7205 [21]	≥380 mm ≥40ϕ	≥8ϕ	test time shall be within 1 to 10 min
CAN/CSA S806 [22]	≥40ϕ	≥5ϕ	stress rate: 250 to 500 MPa/min
CNR–DT 206 [23]	≥100 mm ≥40ϕ	≥8ϕ	test time shall be within 1 to 10 min
EN 17129 [24]	≥20ϕ	≥50 mm ≥3ϕ	test time shall be within 1 to 5 min
GOST 31938 [25]	≥40ϕ	≥8ϕ	test time shall be within 3 to 10 min
ISO 10406–1 [26]	≥300 mm ≥40ϕ	≥100 mm ≥8ϕ	strain rate: 0.5% to 1.5% per minute, test time shall not exceed 5 min
JSCE–E 531 [27]	≥100 mm ≥40ϕ	≥8ϕ	stress rate: 100 to 500 MPa/min

**Table 4 materials-13-04110-t004:** Tensile strength based on core (a) and effective (b) diameters.

Series	Number of Tests	Tensile Strength						
		AVG [MPa]			SD [MPa]		CV [%]	
		(a)	(b)	(a)/(b)	(a)	(b)	(a)	(b)
A–D8a	7	1164.04	1062.28	9.58%	28.55	26.05	2.45	2.45
A–D8b	5	1175.06	n.a.	n.a.	11.46	n.a.	0.98	0.98
A–D10	5	1320.30	1200.89	9.94%	50.30	45.75	3.81	3.81
A–D12	5	1243.11	1182.51	5.12%	44.93	42.74	3.61	3.61
B–D5a	3	1124.44	1017.52	10.51%	94.77	85.87	8.44	8.44
B–D5b	3	1183.98	1094.82	8.14%	11.28	10.44	0.95	0.95
B–D5c	3	1031.00	990.50	4.09%	29.98	28.80	2.91	2.91
B–D6	3	1193.44	1136.07	5.05%	12.31	11.64	1.02	1.02
B–D8	7	1189.58	1111.31	7.04%	23.94	22.37	2.01	2.01
B–D11	3	1136.30	1095.72	3.70%	60.71	58.54	5.34	5.34
C–D12	3	1415.99	1266.49	11.80%	33.13	29.63	2.34	2.34
D–D12	3	1227.41	1057.33	16.09%	35.13	30.26	2.86	2.86

**Table 5 materials-13-04110-t005:** Modulus of elasticity specification.

Standard	Points at the Load–Strain Curve	
	Start Point	End Point
ASTM D7205 [21]	a strain of 0.001 * 25% of ultimate strain	a strain of 0.003 * 50% of ultimate strain
CAN/CSA S806 [22]	25% of ultimate load	50% of ultimate load
CNR–DT 203 [23] GOST 31938 [24] ISO 10406–1 [25]	20% of ultimate load	50% of ultimate load
EN 17129 [26]	a strain of 0.001	a strain of 0.005
JSCE–E 531 [27]	20% of ultimate load	60% of ultimate load

* for materials that fail at strain below 0.006.

**Table 6 materials-13-04110-t006:** Modulus of elasticity based on core (a) and effective (b) diameter determined according to various standards.

Series	Statistic	ASTM D7205 [21]		CAN/CSA S806 [22] and		CNR–DT 206 [23] GOST 31938 [25] ISO 10406–1 [26]		EN 17129 [24]		JSCE–E 531 [27]	
		(a)	(b)	(a)	(b)	(a)	(b)	(a)	(b)	(a)	(b)
A–D8a	AVG	41.16	37.56	46.39	42.32	46.52	42.45	42.57	38.85	46.17	42.13
	SD	12.83	11.71	2.92	2.64	3.27	2.98	10.11	9.23	2.55	2.33
	CV	31.17	31.17	6.30	6.24	7.02	7.02	23.76	23.75	5.51	5.52
A–D8b	AVG	44.88	n/a	48.66	n/a	49.27	n/a	47.74	n/a	49.68	n/a
	SD	11.18	n/a	1.32	n/a	1.78	n/a	8.12	n/a	0.58	n/a
	CV	24.90	n/a	2.70	n/a	3.62	n/a	17.02	n/a	1.17	n/a
A–D10	AVG	57.45	52.26	57.01	51.85	56.85	51.71	55.96	50.9	56.58	51.46
	SD	15.34	13.95	3.26	2.96	2.45	2.22	10.51	9.57	1.85	1.68
	CV	26.71	26.70	5.71	5.71	4.30	4.30	18.79	18.80	3.27	3.27
A–D12	AVG	54.7	51.84	56.11	53.37	56.69	53.93	55.98	53.25	56.76	54.00
	SD	11.84	11.37	3.46	3.30	3.87	3.68	8.76	8.33	3.46	3.29
	CV	21.64	21.93	6.17	6.18	6.82	6.83	15.65	15.65	6.10	6.09
B–D5a	AVG	44.25	40.03	48.94	44.28	49.15	44.46	47.38	42.85	49.04	44.37
	SD	7.46	6.75	1.43	1.29	1.20	1.09	5.11	4.64	1.48	1.34
	CV	16.86	16.86	2.92	2.92	2.45	2.45	10.79	10.83	3.01	3.02
B–D5b	AVG	53.51	49.48	50.62	46.8	50.37	46.58	52.41	48.47	50.12	46.34
	SD	3.33	3.08	1.04	0.96	0.89	0.82	2.57	2.37	0.94	0.87
	CV	6.22	6.22	2.05	2.06	1.76	1.76	4.90	4.90	1.87	1.88
B–D5c	AVG	50.81	48.81	52.35	50.3	52.19	50.14	51.44	49.42	52.4	50.34
	SD	2.87	2.76	0.38	0.37	1.64	1.57	1.94	1.86	1.65	1.58
	CV	5.66	5.65	0.73	0.73	3.14	3.13	3.77	3.77	3.15	3.15
B–D6	AVG	52.07	49.58	49.87	47.48	49.88	47.49	50.82	48.39	51.11	48.67
	SD	9.57	9.12	1.05	0.99	0.72	0.69	6.56	6.25	2.79	2.66
	CV	18.39	18.39	2.10	2.09	1.44	1.45	12.91	12.91	5.47	5.47
B–D8	AVG	51.12	49.09	52.7	49.24	52.97	49.49	53.59	50.06	52.83	49.36
	SD	6.22	4.41	0.68	0.64	0.83	0.77	3.04	2.84	0.69	0.65
	CV	12.18	8.98	1.30	1.29	1.56	1.56	5.67	5.67	1.31	1.32
B–D11	AVG	56.01	54.01	51.79	49.94	52.40	50.53	55.58	53.61	52.91	51.02
	SD	2.27	2.20	1.10	1.06	0.94	0.90	1.66	1.61	1.89	1.82
	CV	4.06	4.06	2.13	2.12	1.79	1.78	2.99	3.00	3.57	3.57
C–D12	AVG	68.02	60.84	65.04	58.18	65.57	58.65	67.82	59.76	65.18	58.30
	SD	5.28	4.72	1.89	1.69	0.95	0.84	2.95	4.01	0.95	0.85
	CV	7.76	7.76	2.90	2.91	1.44	1.44	4.35	6.71	1.46	1.47
D–D12	AVG	62.41	53.76	56.76	48.89	57.52	49.55	63.39	54.60	57.53	49.56
	SD	2.83	2.43	1.37	1.18	1.66	1.43	1.50	1.30	1.12	0.97
	CV	4.53	4.52	2.41	2.40	2.89	2.89	2.37	2.37	1.95	1.95

**Table 7 materials-13-04110-t007:** Ultimate strains determined by two different methods.

Series	Number of Tests	Ultimate Strain						MI/MII
		Method I			Method II			
		AVG [%]	SD [%]	CV [%]	AVG [%]	SD [%]	CV [%]	ΔAVG [%]
A–D8a	7	2.46	0.41	16.54	2.52	0.16	6.41	–2.38
A–D8b	5	2.37	0.20	8.31	2.42	0.06	2.37	–2.07
A–D10	5	2.32	0.21	9.16	2.32	0.14	6.32	0.00
A–D12	5	2.28	0.15	6.68	2.22	0.09	3.89	2.70
B–D5a	3	2.37	0.21	8.67	2.30	0.14	6.02	3.04
B–D5b	3	2.38	0.07	2.94	2.34	0.07	3.02	1.71
B–D5c	3	2.03	0.06	3.17	1.97	0.07	3.64	3.05
B–D6	3	2.32 *	0.17 *	7.31	2.39	0.07	2.97	–2.93
B–D8	7	2.30	0.07	3.11	2.26	0.05	2.24	1.77
B–D11	3	2.14	0.07	3.06	2.19	0.09	3.97	–2.28
C–D12	3	2.23	0.02	0.93	2.18	0.03	1.50	2.29
D–D12	3	2.06	0.02	0.84	2.16	0.11	4.98	–4.63

* only two specimens were considered in analysis due to extensometer fault.

**Table 8 materials-13-04110-t008:** Comparison of tensile properties of different type of bars.

Type (Series)	Tensile Strength [MPa]			Modulus of Elasticity [GPa]			Ultimate Strain [%]		
	AVG	SD	CV	AVG	SD	CV	AVG	SD	CV
A (A–D12)	1243.11	44.93	3.61	56.69	3.86	6.83	2.20	0.10	4.44
B (B–D11)	1136.30	60.71	5.34	52.40	0.94	1.79	2.17	0.09	4.04
C (C–D12)	1415.99	33.13	2.34	65.57	0.95	1.44	2.16	0.03	1.52
D (D–D12)	1227.41	35.13	2.86	57.52	1.66	2.89	2.14	0.11	5.30

**Table 9 materials-13-04110-t009:** Comparison of tensile properties of bars produced with the different process.

Factor	Series	Tensile Strength [MPa]			Modulus of Elasticity [GPa]			Ultimate Strain [%]		
		AVG	SD	CV	AVG	SD	CV	AVG	SD	CV
Homogeneity of batch	A–D8a	1164.04	28.55	2.45	46.39	2.92	6.30	2.52	0.16	6.41
	A–D8b	1175.06	11.46	0.98	48.66	1.32	2.70	2.42	0.06	2.37
Resin modification	B–D5a	1124.44	85.87	8.44	48.94	1.43	2.92	2.30	0.14	6.02
	B–D5b	1183.98	11.28	0.95	50.62	1.04	2.05	2.34	0.07	3.02
	B–D5c	1031.00	29.98	2.91	52.35	0.38	0.73	1.97	0.07	3.64

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
