# Peer review of "Comparison of Tensile Properties of Glass Fibre Reinforced Polymer Rebars by Testing According to Various Standards"

_materials, 2020, doi:10.3390/ma13184110_

Round 1

Reviewer 1 Report

Dear authors,

Please find my response as follows:

The manuscript » Comparison of Tensile Properties of Glass Fibre Reinforced Polymer Rebars by Testing According to Various Standards « submitted to the Journal “Materials” is prepared in appropriate format, but minor revisions should be done:

1) Line 236 Table 4: The authors should explain why different number of tests were performed at each sample/method.

2) Line 247 Table 5: At the Table, standard methods are listed. The same as start and end point. Why there are no details regarding start and end point at standards CNR-DT 203 and ISO 10406-1?

3) Do you have an information how glass fibres are distributed in the bars? Maybe SEM analysis/images would be important to see, how fibres are distributed in epoxy and vinyl ester. Namely this also effects the tensile results.  

4) Listed literature is quite old. Authors should add newly published results and analysis regarding this topic.

Reviewer 2 Report

The presented work is good, however, following comments should be addressed:

  1. Add relevant latest literature review in introduction section.
  2. Add a brief section before conclusions for possible implementation of this study in construction industry.
  3. Compare the revealed trends of this work with some past relevant studies. Also, comparison with steel rebars (used widely in construction industry) is good.
  4. Avoid short paragraphs, e.g. lines 320-322 is too short.
  5. Graphs of Figure 4 should be on one page. Avoid splitting one figure on two pages.

Reviewer 3 Report

The study is very interesting considering all the mechanical characterizations, diameter measurement methods, standards, and bar types taken into account.

The article is clear, well organized and the experiments are very complete and carried out rigorously.

However, I have some recommendations listed below:

- Page 3, line 114: How many diameter measurements were done on each bar?

- Table 2: Measurements error should be added.

- Page 3, line 135: A picture of the special anchorage could help the understanding.
